# Editing *eIF4E* in the Watermelon Genome Using CRISPR/Cas9 Technology Confers Resistance to ZYMV

**DOI:** 10.3390/ijms252111468

**Published:** 2024-10-25

**Authors:** Maoying Li, Yanhong Qiu, Dongyang Zhu, Xiulan Xu, Shouwei Tian, Jinfang Wang, Yongtao Yu, Yi Ren, Guoyi Gong, Haiying Zhang, Yong Xu, Jie Zhang

**Affiliations:** 1Beijing Vegetable Research Center, Beijing Academy of Agricultural and Forestry Sciences, Beijing 100097, China; qiuyanhong@nercv.org (Y.Q.); zhudongyang0618@163.com (D.Z.); xuxiulan@nercv.org (X.X.); wangjinfang@nercv.org (J.W.); yuyongtao@nercv.org (Y.Y.); renyi@nercv.org (Y.R.); gongguoyi@nercv.org (G.G.); zhanghaiying@nercv.org (H.Z.); xuyong@nercv.org (Y.X.); 2College of Horticulture, Hunan Agricultural University, Changsha 410128, China

**Keywords:** watermelon, eukaryotic translation initiation factor 4E (eIF4E), CRISPR/Cas9

## Abstract

Watermelon is one of the most important cucurbit crops, but its production is seriously affected by viral infections. Although eIF4E proteins have emerged as the major mediators of the resistance to viral infections, the mechanism underlying the contributions of eIF4E to watermelon disease resistance remains unclear. In this study, three *CleIF4E* genes and one *CleIF(iso)4E* gene were identified in the watermelon genome. Among these genes, *CleIF4E1* was most similar to other known *eIF4E* genes. To investigate the role of *CleIF4E1*, CRISPR/Cas9 technology was used to knock out *CleIF4E1* in watermelon. One selected mutant line had an 86 bp deletion that resulted in a frame-shift and the expression of a truncated protein. The homozygous mutant exhibits developmental defects in plant growth, leaf morphology and reduced yield. Furthermore, the mutant was protected against the zucchini yellow mosaic virus, but not the cucumber green mottled mosaic virus. In summary, this study preliminarily clarified the functions of eIF4E proteins in watermelon. The generated data will be useful for elucidating eIF4E-related disease resistance mechanisms in watermelon. The tissue-specific editing of *CleIF4E1* in future studies may help to prevent adverse changes to watermelon fertility.

## 1. Introduction

Potyviruses, which are the largest group of plant RNA viruses, include many agriculturally important viruses [1,2]. They can infect a wide range of plant species and are a major threat to global agricultural production. Potyviruses usually contain a large positive-sense single-stranded RNA genome that encodes a long polyprotein, which is proteolytically hydrolyzed to individual mature peptides [3,4].

Because of their limited coding capacity, viruses usually employ host cellular factors to complete their infection cycle. The genomic RNA of potyviruses lacks a 5′ cap. Instead, these viruses have evolved to produce a viral genome-linked protein (VPg) with an analogous function to the mRNA cap [5]. Specifically, VPg covalently links the 5′ end of the viral RNA and interacts with the host eukaryotic translation initiation factor 4E (eIF4E) family proteins to initiate translation [6,7]. Thus, mutating *eIF4E* genes can alter this interaction and prevent the virus from exploiting the host cellular machinery, thereby protecting the host from infection (i.e., recessive resistance) [8]. To date, *eIF4E* family genes have emerged as major recessive factors for many viruses, especially potyviruses [5,7,9,10]. Durable and effective resistance due to the inactivation of eIF4E has been achieved in the model plant Arabidopsis [11] as well as in a wide range of crops, including tomato [12,13,14], Chinese cabbage [15,16], tobacco [17,18], wheat [19,20], potato [21] and beet [22]. As the eIF4E factors are essential for the host endogenous translation process, inactivating eIF4E may differentially affect diverse plants [7]. In cucumber, disrupting *eIF4E* expression using CRISPR/Cas9 technology does not result in observable alterations to plant development, but the generated plants exhibit broad-spectrum resistance to many viruses (e.g., ZYMV, papaya ring spot mosaic virus-W, cucumber vein yellowing virus and watermelon mosaic virus) [23,24]. This reflects the potential utility of *eIF4E* genes for developing disease-resistant cultivars. However, inactivating *eIF4E* genes in *Cucumis melo* has also been associated with obvious growth defects or abnormal plant fertility [25], which may limit the use of this gene for improving agricultural production.

In plants, the small *eIF4E* family includes genes encoding eIF4E and its isoform eIF(iso)4E. These genes may have evolved independently in distinct plant species, but they are highly responsive to different potyvirus infections [7]. In most plants, such as cucumber [23] and tobacco [18], a single *eIF4E* allele is generally associated with the susceptibility to certain potyviruses, whereas in some plants, eIF(iso)4E mediates the resistance to viral infections. Examples include Chinese cabbage [16] infected with the turnip mosaic virus and beet infected with the beet mild yellowing virus [22]. Additionally, two or more *eIF4E* family genes may cooperatively mediate the resistance to potyviruses [26,27,28]. In Arabidopsis, simultaneously mutating the genes encoding eIF4E and eIF(iso)4E leads to a more durable resistance to potyviruses than mutating only the gene encoding eIF(iso)4E [29,30].

Watermelon (*Citrullus lanatus*) is an important cucurbit crop because its fruit is consumed globally [31]. However, during cultivation, watermelon plants are susceptible to many viruses, especially potyviruses [31]. ZYMV, which is a typical potyvirus, is the causative agent of a destructive disease of cucurbit crops. It seriously affects crop quality and yield, resulting in considerable economic losses worldwide [32]. Selecting or breeding resistant cultivars is the most viable strategy for protecting crops from viral infections, thereby minimizing economic losses.

In this study, the watermelon genome was screened to identify *eIF4E* family members. Additionally, the main *CleIF4E* gene was edited by CRISPR/Cas9 technology and non-transgenic homozygous mutants (T_2_ generation) were obtained. This mutant not only exhibits defects on plant morphology, but also on male fertility ability. Furthermore, the mutant was resistant to infections of zucchini yellow mosaic virus (ZYMV), but not the cucumber green mottled mosaic virus (CGMMV). The roles of other *CleIF4E* genes were also explored in *CleIF4E1* mutants and viral-infected mutants, revealing that *CleIF4E1* plays a critical role in plant growth and development and also viral resistance.

## 2. Results

### 2.1. Identification of CleIF4E Genes in Watermelon

Amino acid sequences of known eIF4E and eIF(iso)4E proteins were downloaded and used as queries for a BLASTP search of the watermelon genome to identify potential homologs. Three copies of *eIF4E* genes (*CleIF4E1*, *CleIF4E2* and *CleIF4E3*) and one copy of an *eIF(iso)4E* gene (*CleIF(iso)4E*) were identified (Table 1). Other relevant details regarding these genes (e.g., name, sequence length and genome locus) and the encoded proteins (e.g., number of amino acids, molecular weight and pI) are provided in Table 1.

To investigate the evolutionary relationships of CleIF4E proteins, we used a neighbor-joining method to construct a phylogenetic tree for eIF4E proteins from watermelon, Arabidopsis and other identified plant species. In the phylogenetic tree, these proteins were divided into three subgroups, among which subgroup I was the dominant group. Subgroup I CleIF4E1 proteins were similar to AteIF4E1 from Arabidopsis (Figure 1).

### 2.2. Generation of Genome-Edited Plants Using Crispr/Cas9 Technology

Earlier research indicated that eIF4E proteins are required for the host endogenous translation process; deleting the corresponding genes could lead to unexpected developmental defects or even lethality [7,25,33]. Thus, we first designed sgRNAs that specifically target *CleIF4E1*, but not *CleIF4E2* or *CleIF4E3*, to ensure only *CleIF4E1* was knocked out using CRISPR/Cas9 technology. Two sgRNAs targeting the exon of *CleIF4E1* (Figure 2B) with high computational prediction scores were selected. These sgRNAs were cloned into CRISPR/Cas9 vectors, which were then inserted into *Agrobacterium tumefaciens* cells for the subsequent transformation of cotyledons (Figure 2A). Putative T_0_ mutant transgenic lines were generated and confirmed by Sanger sequencing before they were grown in soil. Next, Cas9-free watermelon lines were selected from the T_1_ generation plants using *Cas9* gene-specific primers and then the sgRNA target region was amplified by PCR. An analysis of one of the Cas9-free lines detected different mixed peaks, and this line was grown in a greenhouse and self-pollinated, after which its seeds were harvested for the selection of homozygous mutant lines. After the T_2_ seeding, we identified one mutant with an 86 bp deletion (position 153 to 239) in the *CleIF4E1* mRNA sequence (Figure 2C). This deletion resulted in a frame-shift in the coding region and the incomplete expression of CleIF4E1 (Figure 2D).

### 2.3. Effect of Knocking Out CleIF4E1 Using CRISPR/Cas9 Technology

In the current study, we analyzed the effects of knocking out *CleIF4E1* on watermelon plant growth and development. The comparison with wild-type (WT) plants indicated that inactivating *CleIF4E1* in watermelon results in weak developmental defects. At the seedling stage, the mutant exhibited weak growth and the leaves were relatively small and wrinkled (Figure 3A). During the subsequent growth stages, the mutant plant exhibited dwarfism, but the leaf wrinkling symptom gradually weakened (Figure 3A). Moreover, there were changes to reproductive ability. Specifically, the fruit of the mutant line was smaller than that of the WT control (Figure 3C). After harvest, only a few seeds were obtained from the mutant; these seeds were smaller than the WT seeds, which may have contributed to the decrease in the mutant seed weight (Figure 3D–F).

The expression levels of *CleIF4E2*, *CleIF4E3* and *CleIF(iso)4E* were also detected with a quantitative reverse transcription polymerase chain reaction (qRT-PCR) method. Six individual mutant plants were collected from the uppermost leaves for the extraction of total RNA. The results showed that the relative expression levels of the other *eIF4E* family genes did not increase in the *CleIF4E1* knockout mutant (Figure 3B).

### 2.4. Evaluation of the Mutant Responses to ZYMV and CGMMV Infections

An infectious clone of ZYMV fused with eGFP was used in this study [34]. Watermelon cotyledons were infiltrated with *A. tumefaciens* carrying the infectious clone of ZYMV. At 21 days post-inoculation (dpi), the WT plants had several disease symptoms (e.g., mosaic, yellowing and malformation), whereas the *CleIF4E1* knockout mutant had no detectable disease symptoms (Figure 4A). The uppermost systemic leaves from the ZYMV-infected WT and *CleIF4E1* knockout plants were collected for a Western blot analysis, which revealed ZYMV accumulated much less in the *CleIF4E1* knockout mutant leaf than in the WT leaf (Figure 4B). An RT-PCR analysis was also conducted to clarify the ZYMV RNA expression profiles. Six and two samples were obtained for the WT and *CleIF4E1* knockout plants, respectively. The expected band was clearly detected for the WT control, which was in contrast to the very weak band detected for the *CleIF4E1* knockout mutant (Figure 4C).

According to these results, CleIF4E1 likely plays a dominant role during a ZYMV infection. Thus, knocking out *CleIF4E1* may help to protect watermelon from ZYMV. The success of the potyvirus infections of most plants depends on eIF4E. Interestingly, other *eIF4E* paralogs may be recruited by viruses if the main *eIF4E* gene is knocked out [9]. In some plants, an *eIF(iso)4E* gene rather than an *eIF4E* gene contributes to defense responses to certain viral infections [16,22,35]. Accordingly, we attempted to identify the *CleIF4E* genes induced during the infection by ZYMV. However, *CleIF4E2*, *CleIF4E3* and *CleIF(iso)4E* transcript levels did not differ significantly between the WT and *CleIF4E1* knockout plants infected with ZYMV (Figure 4D,E). This indicates that potyviruses may compete with host plants for the binding to eIF4E, thereby altering plant processes.

CGMMV, which belongs to the genus *Tobamovirus*, is also one of the most economically important viruses that threaten global cucurbit crop production [36]. There are currently no watermelon species that are fully resistant to CGMMV, but watermelon PI595203, which exhibits eIF4E-mediated resistance to ZYMV [37,38], is reportedly tolerant to CGMMV [39]. In the present study, we evaluated the susceptibility of the *CleIF4E1* knockout mutant to CGMMV. Briefly, mutant and WT leaves were inoculated with sap from CGMMV-infected plants. At 21 dpi, bubbling and mosaic symptoms were detected on the WT and mutant plants (Figure 4F). Three samples were collected from CGMMV-infected WT and *CleIF4E1* knockout plants. An RT-PCR analysis involving CGMMV-specific primers was performed to examine viral accumulation. The results confirmed that CGMMV was able to infect the WT and *CleIF4E1* knockout plants (Figure 4G), implying knocking out *CleIF4E1* did not alter the susceptibility of watermelon to CGMMV.

## 3. Discussion

In eukaryotic organisms, eIF4E proteins are the key regulators of the translation process. These proteins can bind to the 5′-m7G cap of mRNA to form a complex with other translation initiation factors and initiate protein synthesis [7,40]. Viruses in various families are limited regarding their coding capacity and usually hijack host factors to maintain the infection cycle. Notably, VPg produced by potyviruses may mimic the plant cap structure to facilitate the interaction with host eIF4E proteins necessary for viral multiplication. Hence, inactivating eIF4E proteins in various plants may prevent the virus from using the host cellular machinery, resulting in genetic resistance to viral infections. In fact, varieties with durable and broad-spectrum disease resistance have been developed via genetic modifications to *eIF4E* or its isoforms [7]. Because eIF4E proteins also play indispensable roles during the translation of plant mRNA, silencing *eIF4E* may result in diverse effects on plant development in various crops [7,23,25]. Inactivating a single *eIF4E* gene usually leads to very weak or no changes to plant development [20,23,28]. However, knocking out *eIF4E* in *Cucumis melo* reportedly leads to male sterility [25], which may limit the use of eIF4E in agriculture.

Watermelon is a valuable cucurbit crop that is frequently infected by many viruses, including potyviruses. Unfortunately, there are a limited number of disease-resistant genes and watermelon cultivars that are resistant to viral infections remain rare. Although eIF4E has been identified as a major factor in increasing plant susceptibility to diverse potyviruses, eIF4E functions in watermelon have not been thoroughly investigated. To the best of our knowledge, this is the first study to comprehensively analyze genes encoding eIF4E and eIF(iso)4E in the watermelon genome. Three copies of *CleIF4E* genes and one copy of *CleIF(iso)4E* were identified (Table 1). Of the encoded proteins, CleIF4E1 was highly homologous to previously characterized eIF4E proteins. Next, *CleIF4E1* was knocked out using CRISPR/Cas9 technology. We ultimately obtained a homozygous mutant that contained an 86 bp deletion in the *CleIF4E1* coding region, leading to a frame-shift in the coding region and the premature termination of CleIF4E1 expression (Figure 2C,D). However, knocking out *CleIF4E1* resulted in developmental defects in watermelon plants. More specifically, the *CleIF4E1* knockout mutant grew slightly slower than the WT control and had curled leaves. In addition, the reproductive capacity of the mutant was also affected. Pollen production decreased markedly in the mutant plants, leading to decreases in seed yield, size and weight (Figure 3). These results confirmed that the eIF4E not only participated in the plant growth but was also involved in plant fertility, which has been found in melon [25]. Except the eIF4E, other translation initiation factors have also been found to affect the reproduction of plants [41]. Whether the eIF4E is directly involved in the reproductive process of plants or influences other transcription factors functions need to be further discussed.

Because eIF4E proteins are the main factors mediating the susceptibility to potyviruses, we assessed the ability of the *CleIF4E1* knockout mutant to tolerate a potyvirus infection. The mutant was protected against ZYMV (potyvirus), but not against CGMMV (belonging to the genus *Tobamovirus*) (Figure 4). The study findings imply eIF4E proteins have a wide range of functions during the viral life cycle. Moreover, the viral resistance mediated by these proteins may not be limited to potyviruses, with protective effects against various viruses from different genera [7]. Thus, whether the *CleIF4E1* knockout mutant is tolerant or resistant to other viral infections must be explored.

In many plants, *eIF4E* family members have redundant functions. Hence, if one *eIF4E* gene is mutated or its expression is disrupted, other *eIF4E* genes will be recruited to maintain normal plant cellular mechanisms. For example, in tomato, knocking out the primary gene related to susceptibility (*eIF4E1*) causes viruses to recruit their paralog (*eIF4E2*) [28]. The *eIF(iso)4E* gene in some plants plays a major role during the resistance-related response to certain viral infections [16,22]. However, the expression levels of *CleIF4E2*, *CleIF4E3* and *CleIF(iso)4E* were not influenced by the lack of *CleIF4E1* expression in the current study. Following the infection by ZYMV, we did not detect the induced expression of *CleIF4E2*, *CleIF4E3* and *CleIF(iso)4E* in the *CleIF4E1* knockout mutant. Moreover, *CleIF4E1* expression was not up-regulated during the ZYMV infection of WT plants. It is possible that viruses may compete with the host for eIF4E, thereby affecting host activities.

In summary, we identified *CleIF4E* and *CleIF(iso)4E* genes in watermelon and explored the functions of these genes on the basis of CRISPR/Cas9 technology and resistance to ZYMV and CGMMV. The functional annotation of *eIF4E* genes in watermelon in terms of their roles in viral resistance is critical for breeding disease-resistant varieties. As *eIF4E* mutations cause adverse effects on fertility in watermelon, the tissue-specific editing of these genes using a precise CRISPR-based technique may mitigate this influence. In addition, other reported resistant genes should be noted, including the ribosome-inactivating protein (RIP) genes found in watermelon [42] and the eukaryotic translation initiation factor and other interacting factors, such as EXA1 in *Arabidopsis thaliana* [43].

## 4. Materials and Methods

### 4.1. Identification of eIF4E Genes in the Watermelon Genome

To identify the genes encoding eIF4E and eIF(iso)4E proteins in watermelon, amino acid sequences of eIF4E family proteins in the National Center for Biotechnology Information (NCBI) database were downloaded and used to conduct a BLASTP analysis of the *Citrullus lanatus* 97103 (v2.0) genome in the *Cucurbitaceae* database. Three candidate genes were identified and the conserved domains were confirmed on the basis of analyses involving CDD (https://www.ncbi.nlm.nih.gov/cdd/; accessed on 18 June 2023) and InterPro (https://www.ebi.ac.uk/interpro/; accessed on 18 June 2023) databases. A phylogenetic tree for CleIF4E proteins was constructed using the neighbor-joining method (1000 bootstrap replicates) of MEGA (v7.0) and visualized using the Evolview 3.0 online program (https://www.evolgenius.info/evolview/, accessed on 18 May 2023).

### 4.2. RNA Extraction and qRT-PCR Assays

The expression levels of the *eIF4E* genes were analyzed via qRT-PCR, which was completed with a One Step SYBR^®^ PrimeScript™ RT-PCR Kit (Takara, Beijing, China) using an Applied Biosystems 7500 System (Thermo Scientific, Beijing, China). During the qRT-PCR test, three technical replicates were analyzed for individual samples and the mean quantification cycle value of the triplicate reactions was used to calculate the relative expression levels of the target genes according to the 2^−ΔΔCt^ method. A gene encoding the clathrin adaptor complex subunit was used as an internal standard [44]. The significant differences in the relative expression of the target genes between the treated groups and control group were analyzed by the software graphpad prism (https://www.graphpad.com/features accessed on 18 June 2023) using Student’s *t*-test method; these differences were judged by the p value (ns, *p* > 0.05; ****, *p* < 0.001). The data presented are the mean ± SD of three biological independent experiments. Details regarding the primer pairs are listed in Appendix A.

### 4.3. Genetic Transformation of Watermelon

To generate *CleIF4E1* knockout mutant plants, CRISPR RGEN tools (http://www.rgeneome.net, accessed on 20 October 2021) were used to find sgRNA target sequences. Two sgRNAs were designed for *CleIF4E1*. The sgRNAs were cloned into a vector containing the *Cas9* gene under the control of the CaMV 35S promoter. The recombinant vector was inserted into *A. tumefaciens* cells for the subsequent infiltration of watermelon plants [45]. After shoots and then roots were induced, we screened the T_0_ plants via a PCR using target gene-specific primers (Appendix A). Seeds were collected from putative transformants for T_1_ heteroduplex analyses. T_2_ plants were selected to generate homozygous mutants, which were verified by amplifying sgRNA target sequences by PCR. All destination vectors and the primers are listed in Appendix A. All plasmids were confirmed by Sanger sequencing.

### 4.4. Virus Inoculations and Evaluation of Resistance

The ZYMV-GFP infectious clone with the 35S promoter was kindly provided by the Gu Qin-sheng laboratory (Zhengzhou Fruit Research Institute, Chinese Academy of Agricultural Sciences). The infectious clone was transformed into *A. tumefaciens* strain EHA105 cells, which were further cultured in LB medium to an optical density of 0.6 tested at 600 nm. The cultures were collected and then suspended with MMA buffer (10 mM MES/NaOH, pH 5.6, 10 mM MgCl_2_, 100 mM acetosyringone), which were subsequently injected into WT and *CleIF4E1* knockout mutant plants at the first-leaf stage. The treated watermelon plants were grown in a greenhouse on a cycle of 12 h light at 30 °C and 12 h dark at 20 °C.

In addition, CGMMV stored at −80 °C was immediately homogenized with 0.1 M potassium phosphate buffer (pH 7.0) for the viral inoculum, which was then mechanically inoculated with Carborundum powder onto the first true leaves of the watermelon plants. The symptoms were photographed using a digital camera at 21 dpi.

### 4.5. Virus Detection

To detect ZYMV, a Western blot analysis was performed using an anti-GFP antibody. Briefly, leaves of three watermelon plants from each treatment were collected as one sample and immediately ground in liquid nitrogen for a sodium dodecyl sulfate polyacrylamide gel electrophoresis analysis using 2× loading buffer. The gel-separated proteins were transferred to a polyvinylidene difluoride (PVDF) membrane (GE Healthcare, Beijing, China) and incubated with an anti-GFP monoclonal antibody (Cell Signaling Technology, Beijing, China). After incubating with a horseradish peroxidase-labeled secondary antibody (Sigma-Aldrich, Beijing, China), the signals were visualized by treating the PVDF membrane with the Pierce™ ECL Western Blotting Substrate (Thermo Scientific, Beijing, China). Plant actin, which was used as the internal control, was detected with a polyclonal antibody (Sigma-Aldrich, Beijing, China). Furthermore, a PCR method was used to detect ZYMV and CGMMV; the primer pairs are listed in Appendix A. Briefly, the uppermost systemic leaves were collected from three virus-infected plants or the mock-inoculated plants as one sample and then ground to extract the total RNA as described above. These biological experiments were repeated for tree times.

## 5. Conclusions

In this study, four *eIF4E* family members were identified in the watermelon genome and the *CleIF4E1* gene was knocked out using CRISPR/Cas9 technology. The resulting non-transgenic mutants (T_2_ generation) exhibit developmental defects in plant growth and fertility. Furthermore, the mutant was protected against the zucchini yellow mosaic virus, but not the cucumber green mottled mosaic virus. The study findings may provide the basis for future research conducted to generate viral disease-resistant watermelon cultivars.

## Figures and Tables

**Figure 1 ijms-25-11468-f001:**
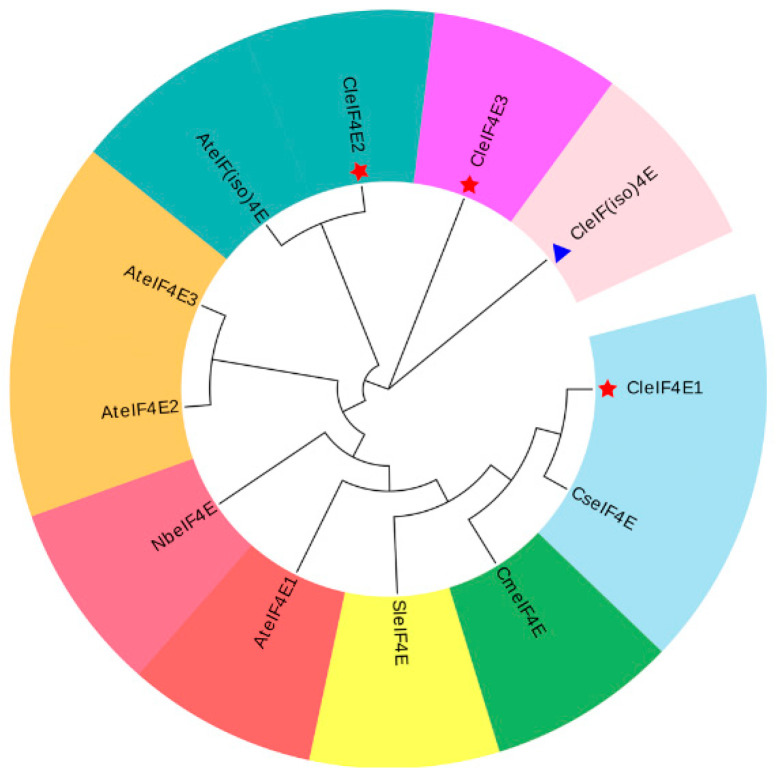
Phylogenetic relationships among CleIF4E proteins and other eIF4E proteins from various plants in the NCBI database. The red pentagram symbol represents the three identified CleIF4E proteins, while the blue triangle symbol represents one identified CleIF(iso)4E protein.

**Figure 2 ijms-25-11468-f002:**
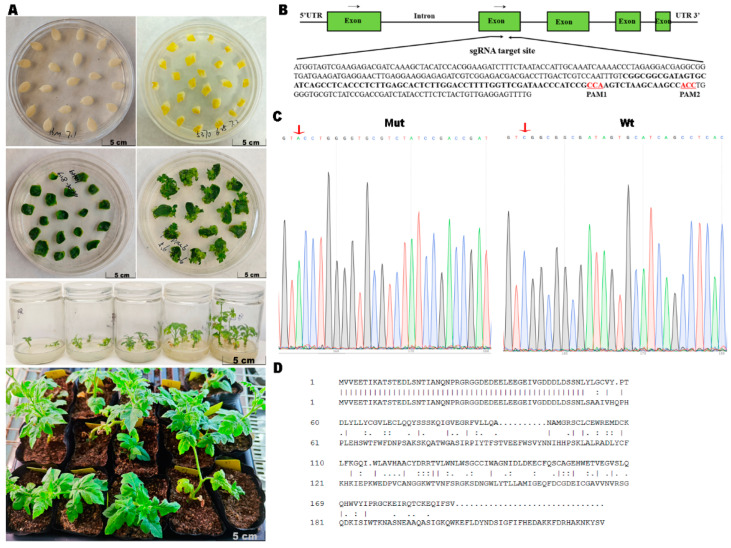
Editing of *CleIF4E1* in watermelon using CRISPR/Cas9 technology. (**A**) Process for selecting edited lines from the cotyledon to adult plants. (**B**) Genomic map and *CleIF4E1* target sites. The protospacer adjacent motif (PAM) is indicated with underlined red letters. (**C**,**D**) Comparison of the eIF4E nucleotide sequence and amino acid sequence between wild-type (wt) and mutant lines. Red arrows indicate the Cas9 cleavage sites.

**Figure 3 ijms-25-11468-f003:**
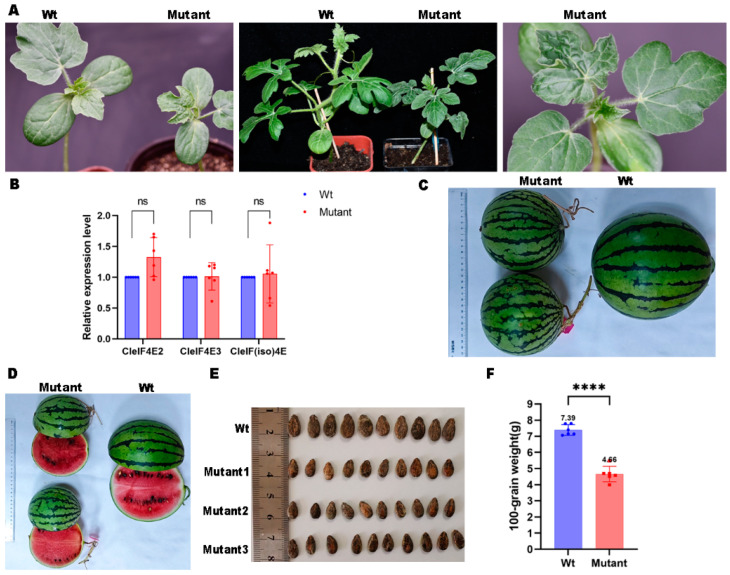
The effect of inactivating *CleIF4E1* using CRISPR/Cas9 technology on watermelon plant growth and development. (**A**) Phenotypes of the *CleIF4E1* knockout mutant at different growth stages and wrinkled and curled mutant leaves. (**B**) Relative expression levels of other *CleIF4E* genes in the *CleIF4E1* knockout mutant. The results were shown as means ± standard deviations of three independent experiments. The ns means no significant differences, while **** means *p* < 0.001. (**C**) Compared with the wild-type (Wt) watermelon fruit, the mutant fruit was significantly smaller. (**D**–**F**) Seeds from the mutant plants were fewer, noticeably smaller and had a significantly lower hundred-seed weight than the Wt plant. Mutants 1–3 represent three individual mutant fruits.

**Figure 4 ijms-25-11468-f004:**
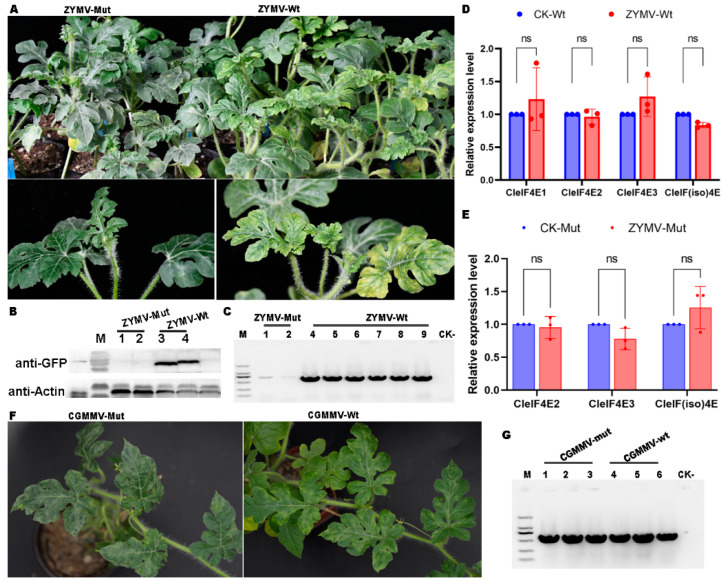
Disease symptoms on *CleIF4E1* knockout mutant and wild-type (Wt) plants infected with ZYMV and CGMMV. (**A**) Symptoms on ZYMV-GFP-infected mutant and Wt plants at 21 days post-inoculation (dpi). (**B**) Accumulation of ZYMV-GFP in mutant and Wt plants determined using an anti-GFP antibody. (**C**) RT-PCR results for the viral accumulation in mutant and Wt plants inoculated with ZYMV-GFP. (**D**,**E**) Relative *CleIF4E* expression levels in mutant and Wt plants during an infection by ZYMV-GFP. The results were shown as means ± standard deviations of three independent experiments. The ns means no significant differences. (**F**) Symptoms on mutant and Wt plants infected with CGMMV at 21 dpi. (**G**) RT-PCR results for the viral accumulation in mutant and Wt plants inoculated with CGMMV.

**Table 1 ijms-25-11468-t001:** Details regarding four *eIF4E* genes identified in the watermelon genome.

Gene Name	Gene ID	CDS Length (bp)	Protein Length (aa)	Molecular Weight
CleIF4E1	Cla97C03G058500	708	235	26,537.33
CleIF4E2	Cla97C06G116990	804	267	30,284.48
CleIF4E3	Cla97C10G196220	2253	750	84,106.20
CleIF(iso)4E	Cla97C03G063210	2685	894	101,536.21

## Data Availability

Data are contained within this article and Appendix A.

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
