# Peer review of "Editing eIF4E in the Watermelon Genome Using CRISPR/Cas9 Technology Confers Resistance to ZYMV"

_ijms, 2024, doi:10.3390/ijms252111468_

Round 1
Reviewer 1 Report
Comments and Suggestions for Authors
Manuscript shows the effect of mutanting eIF(iso) 4E on potyvirus resistance and possible phenotypic penalties. Shows data from only one regenerated mutant line, The manuscript can be improved. Editing for English is necessary. The results section has many instances where the authors are referring to other published reports. Thee are other areas where a better explaination of methods and observation are in order. These sections should be moved to the discussion. Please see attached PDF with comments.

Needs to be proofread for noun-verb agreement and overall sentence structure.
Author Response
For research article
Response to Reviewer 1 Comments
|
||
1. Summary |
|
|
Thank you for arranging a timely review of our manuscript. We have carefully considered the reviewers’ critical comments and modified the manuscript according to the reviewers’ thoughtful suggestions. A new version of our manuscript was also submitted. Please find the detailed responses below. Reviewer comments are pasted with bold font and we respond to each point made during the reviews in blue font. All the changes in the revised manuscript were highlighted in red.
|
||
2. Questions for General Evaluation |
Reviewer’s Evaluation |
Response and Revisions |
Does the introduction provide sufficient background and include all relevant references? |
Yes/Can be improved/Must be improved/Not applicable |
Thank you very much for the suggestion. All the point-to-point response were listed in detail as below |
Are all the cited references relevant to the research? |
Yes/Can be improved/Must be improved/Not applicable |
|
Is the research design appropriate? |
Yes/Can be improved/Must be improved/Not applicable |
|
Are the methods adequately described? |
Yes/Can be improved/Must be improved/Not applicable |
|
Are the results clearly presented? |
Yes/Can be improved/Must be improved/Not applicable |
|
Are the conclusions supported by the results? |
Yes/Can be improved/Must be improved/Not applicable |
|
3. Point-by-point response to Comments and Suggestions for Authors |
||
Comments 1: Manuscript shows the effect of mutanting eIF(iso) 4E on potyvirus resistance and possible phenotypic penalties. ——Shows data from only one regenerated mutant line, |
||
Response: Thank you for pointing this out. During the selecting of positive editing plants with PCR method, we found that many mutants belongs to the same type that with many base deletion of the CleIF4E1 genes. So we selected the mutant with the largest deletion of the CleIF4E1 genes for further research. ——The manuscript can be improved. Editing for English is necessary. Response: Thanks for your suggestions. We have carefully revised the manuscript and also edited the English with the help of editing service. Certification is attached. ——The results section has many instances where the authors are referring to other published reports. There are other areas where a better explaination of methods and observation are in order. These sections should be moved to the discussion. Response: Thank you for pointing this out. We have checked this instances and rewritten these parts, some instances were moved to the introduction or discussion part and also given a better explanation. ——Please see attached PDF with comments. Response: Thanks for your suggestions. We respond to each point made in the revised manuscript were made in red font. ——Point 1: Paper has issues with sentence structure, syntax, and verb tense. Manuscript needs to be cleaned up. Response: Thank you for pointing this out. We have revised the manuscript and also improve the English of the manuscript by the language Editing Services. Hope it could fulfill the journal's requirements. ——Point 2: Watermelon is or Watermelons are. Response: Thank you for pointing this out. This sentence was changes as : Watermelon is one of the most important cucurbit crops, but its production is seriously affected by viral infections. ——Point 3: Rewrite sentence: The eIF4E factors have emerged as the major re-sistant genes to defense viral infection. Response: Thank you for pointing this out. This sentence was changes as : Although eIF4E proteins have emerged as the major mediators of the resistance to viral infections, the mechanism underlying the contributions of eIF4E to watermelon disease resistance remains unclear. ——Point 4: unclear Response: Thank you for pointing this out. This sentence was changes as : the mechanism underlying the contributions of eIF4E to watermelon disease resistance remains unclear. ——Point 5: what defect?? Response: Thank you for pointing this out. This sentence was changes as : The homozygous mutant exhibits developmental defects in plant growth, leaf morphology and reduced yield. ——Point 6: suggest Response: Thank you for pointing this out. This sentence was changes as : The tissue-specific editing of CleIF4E1 in future studies may help to prevent adverse changes to watermelon fertility. ——Point 7: Rupp et al. (2019) was first to report in wheat using RNAi. doi: 10.2135/cropsci2018.08.0518 Response: Thank you for pointing this out. We added this reference. wheat [19, 20] 19. Rupp, J.S.; Cruz, L.; Trick, H.N.; Fellers, J.P. RNAi-mediated silencing of endogenous wheat genes EIF(iso)4E2 and EIF4G induce resistance to multiple RNA viruses in transgenic wheat. Crop Science 2019, 59, 2642-2651. [CrossRef] ——Point 7: What does this mean...deeply?? Response: Thank you for pointing this out. This sentence was changes as : Amino acid sequences of known eIF4E and eIF(iso)4E proteins were downloaded and used as queries for a BLASTP search of the watermelon genome to identify potential homologs. ——Point 8: identify other plant species Response: Thank you for pointing this out. This sentence was changes as : To investigate the evolutionary relationships of CleIF4E proteins, we used a neighbor-joining method to construct a phylogenetic tree for eIF4E proteins from watermelon, Arabidopsis, and other identified plant species. ——Point 9: Does this figure/ analysis add anything of importance to the objective of the manuscript? Was this an exhaustive search? define MIF4G PLN and EnvC and iD species. Response: Thank you for pointing this out. This figure was used to prove that the eIF4E factors in watermelon encode a conserved domain (A) and have different motifs (B). As this part did not play very important roles in the manuscript, we deleted related information in the part of results and only provided that in the Materials and Methods part. The Figure 1 was changed as below: Figure 1. Phylogenetic relationships among CleIF4E proteins and other eIF4E proteins from various plants in the NCBI database. The red pentagram symbol represents the three identified CleIF4E proteins, while the blue triangle symbol represents one identified CleIF(iso)4E protein. Despite their relatively low sequence homology, all of these genes included a sequence encoding an IF4E or MIF4G domain (Figure 1A). The encoded proteins varied in terms of the distribution of conserved motifs, which were predicted using the MEME database (Figure 1B). Additionally, eIF4E protein motifs were identified using the MEME Suite database (https://meme-suite.org/meme/; accessed on 18 June 2023) and TBtools [44]. 44. Chen, C.; Chen, H.; Zhang, Y.; Thomas, H.R.; Frank, M.H.; He, Y.; Xia, R. TBtools: an integrative toolkit developed for interactive analyses of big biological data. Mol.Plant. 2020, 13, 1194-1202. [CrossRef][PubMed] 4.1 Identification of eIF4E genes in the watermelon genome To identify genes encoding eIF4E and eIF(iso)4E proteins in watermelon, amino acid sequences of eIF4E family proteins in the National Center for Biotechnology Information (NCBI) database were downloaded and used to conduct a BLASTP analysis of the Citrullus lanatus 97103 (v2.0) genome in the Cucurbitaceae database. Three candidate genes were identified and the conserved domains was confirmed on the basis of analyses involving CDD (https://www.ncbi.nlm.nih.gov/cdd/; accessed on 18 June 2023) and InterPro (https://www.ebi.ac.uk/interpro/; accessed on 18 June 2023) databases. ——Point 10: Howe can you say usually when you cite 1 reference? Response: Thank you for pointing this out. This sentence was moved into the discussion part and changes as : Because eIF4E proteins also play indispensable roles during the translation of plant mRNA, silencing eIF4E may result in diverse effects on plant development in various crops[7, 23, 25]. 7. Zlobin, N.; Taranov, V. Plant eIF4E isoforms as factors of susceptibility and resistance to potyviruses. Front. Plant Sci. 2023, 14. [CrossRef][PubMed] 23. Callot, C.; Gallois, J.L. Pyramiding resistances based on translation initiation factors in Arabidopsis is impaired by male gametophyte lethality. Plant Signal. Behav. 2014, 9, e27940.[CrossRef] 25. Pechar, G.S.; Donaire, L.; Gosalvez, B.; Garcia-Almodovar, C.; Amelia Sanchez-Pina, M.; Truniger, V.; Aranda, M.A. Editing melon eIF4E associates with virus resistance and male sterility. Plant Biotechnol. J. 2022, 20, 2006-2022. [CrossRef][PubMed] ——Point 11: Include scale bar to show size Response: Thank you for pointing this out. We have added the scale bar in the Figure2. As suggested, we simplified the Figure2 as below:
Figure 2. Editing of CleIF4E1 in watermelon using CRISPR/Cas9 technology. (A) Process for selecting edited lines from the cotyledon to adult plants. (B) Genomic map and CleIF4E1 target sites. The protospacer adjacent motif (PAM) is indicated with underlined red letters. (C and D) Comparison of the eIF4E nucleotide sequence and amino acid sequence between wild-type (wt) and mutant lines. Red arrows indicate the Cas9 cleavage sites. ——Point 12: What does this mean? What are the bands in the gel??? label! Response: Thank you for pointing this out. We amplified the full-length of CleIF4E1 CDS from the T1 generation plants to identify homozygous mutant. As this result was confused, we deleted this part. ——Point 13: rewrite section. Results should include only data your study shows, If you are comparing what is in the literature then this should go in the discussion. Response: Thank you for pointing this out. This section was changed as below: 2.3 Effect of knocking out CleIF4E1 using CRISPR/Cas9 technology In the current study, we analyzed the effects of knocking out CleIF4E1 on watermelon plant growth and development. The comparison with wild-type (WT) plants indicated that inactivating CleIF4E1 in watermelon results in weak developmental defects. At the seedling stage, the mutant exhibited weak growth and the leaves were relatively small and wrinkled (Figure 3A). During the subsequent growth stages, the mutant plant exhibited dwarfism, but the leaf wrinkling symptom gradually weakened (Figure 3A). Moreover, there were changes to reproductive ability. Specifically, the fruit of the mutant line was smaller than that of the WT control (Figure 3C). After harvest, only a few seeds were obtained from the mutant; these seeds were smaller than the WT seeds, which may have contributed to the decrease in the mutant seed weight (Figure 3D, E and F). The expression levels of CleIF4E2, CleIF4E3, and CleIF(iso)4E were also detected with a quantitative reverse transcription polymerase chain reaction (qRT-PCR) method. Six individual mutant plants were collected from the uppermost leaves collected for the extraction of total RNA. After testing, results showed that the relative expression levels of other eIF4E family genes was not increased in the CleIF4E1 knockout mutant(Figure 3B). ——Point 14: use genus and species not just common name Response: Thank you for pointing this out. We changed melon into Cucumis melo and moved this part into the ‘Discussion’ part in the revised manuscript. However, knocking out eIF4E in Cucumis melo reportedly leads to male sterility [25], which may limit the use of eIF4E in agriculture. ——Point 15: what do you mean by lately?? Response: Thank you for pointing this out. We changed lately into During the subsequent growth stages. During the subsequent growth stages, the mutant plant exhibited dwarfism, but the leaf wrinkling symptom gradually weakened (Figure 3A). ——Point 16: sentence structure Response: Thank you for pointing this out. We changed that as below: Moreover, there were changes to reproductive ability. ——Point 17: be specific Response: Thank you for pointing this out. It should be from the mutant, we added that. After harvest, only a few seeds were obtained from the mutant. ——Point 18: This should be in discussion not results. Response: Thank you for pointing this out. We moved these to the discussion part and re-written this section. Detail information has been listed in Point 13. ——Point 19: Where didi mutant 2&3 come from? Did you have multiple mutants?? Response: Thank you for pointing this out. The mutant 2&3 means three individual watermelon fruits from three mutant plants. We added the information in the description of figure 3. (D, E and F) seeds from mutant plants were fewer, noticeably smaller and had a significantly lower hundred-seed weight than from Wt plant. Mutant1-3 represent three individual mutant fruit. ——Point 20: not in results should be in intro or discussion Response: Thank you for pointing this out. We moved this part into the fourth paragraph of ‘Introduction’ part. However, during cultivation, watermelon plants are susceptible to many viruses, especially potyviruses [29]. Such as ZYMV, which is a typical potyvirus, is the causative agent of a destructive disease of cucurbit crops. It seriously affects crop quality and yield, resulting in considerable economic losses worldwide [30]. ——Point 21: what method did you use to look at relative expression? what do the error bars refer to? SD, SE ???? Response: Thank you for pointing this out. The 2−ΔΔCt method was used to calculated the relative expression level of the genes. Data presented are the mean ± SD of three biological experiment that treated with ZYMV independently. We added this information on the revised manuscript. Figure 3. (B) Relative expression levels of other CleIF4E genes in the CleIF4E1 knockout mutant. The results were shown as means ± standard deviations of three independent experi-ments. The ns means no significant differences, while **** means p < 0.001. Figure 4. (D, E) Relative CleIF4E expression levels in mutant and Wt plants during an infection by ZYMV-GFP. The results were shown as means ± standard deviations of three independent experiments. The ns means no significant differences. ——Point 22: methods need to have more detail. include stat analysis. For virus work how many biological reps were used.? Expand on virus inoculation. Response: Thank you for pointing this out. We have added more detail in the methods part, including the stat analysis process in 4.2 and virus inoculation in 4.4. 4.2 RNA extraction and qRT-PCR assays The expression levels of eIF4E genes were analyzed via qRT-PCR, which was completed using a One Step SYBR® PrimeScript™ RT-PCR Kit (Takara, Dalian, China) using an Applied Biosystems 7500 System (Thermo Scientific, Shanghai, China). During the qRT-PCR test, three technical replicates were analyzed for individual samples and the mean quantification cycle value of the triplicate reactions was used to calculate the relative expression levels of the target genes according to the 2−ΔΔCt method. A gene encoding the clathrin adaptor complex subunit as an internal standard [42]. The significant differences of the relative expression of target genes between treated groups and control group was analyzed by software graphpad prism in Student's t-test method, which was judged by the p value (ns, p>0.05; ****, p< 0.001). The data presented are the mean ± SD of three biological independent experiment.Details regarding primer pairs are listed in Table S1. 4.4 Virus inoculations and evaluation of resistance The ZYMV-GFP infectious clone with the 35S promoter was kindly provided by the Gu Qin-sheng laboratory (Zhengzhou Fruit Research Institute, Chinese Academy of Agricultural Sciences). The infectious clone was transformed into A. tumefaciens strain EHA105 cells, which were further cultured in LB medium to an optical density of 0.6 tested at 600nm. The cultures were collected and then suspended with MMA buffer (10 mM MES, pH 5.6, 10 mM MgCl2, 100 mM acetosyringone) which were subsequently injected into WT and CleIF4E1 knockout mutant plants at the first-leaf stage. The treated watermelon plants were grown in a greenhouse on a cycle of 12 h light at 30℃and 12 h dark at 20℃. In addition, CGMMV stored at −80 °C was immediately homogenized in 0.1 M potassium phosphate buffer (pH 7.0) for the viral inoculum, which was then mechanically inoculated with carborundum powder onto the first-true leaves of watermelon plants. The symptoms were photographed using a digital camera (Nikon, Tokyo, Japan) at 21 dpi. ——Point 23: expand or cite reference Response: Thank you for pointing this out.We have added the relative reference. The recombinant vector was inserted into A. tumefaciens cells for the subsequent infiltration of watermelon plants [46]. 46. Tian, S.W.; Jiang, L.J.; Gao, Q.; Zhang, J.; Zong, M.; Zhang, H.Y.; Ren, Y.; Guo, S.G.; Gong, G.Y.; Liu, F.; et al. Efficient CRISPR/Cas9-based gene knockout in watermelon. Plant Cell Reports 2017, 36, 399-406. [CrossRef]
|
||
|
||
|
||
|
||
|
||
|
||
|
||
|

Reviewer 2 Report
Comments and Suggestions for Authors
The aim is to study the resistant mechanisms of the eIF4E genes to combat viral infections in watermelon. To further investigate the role of these genes, the CRISPR-Cas9 technology was applied.
Overall, the work allows to improve our knowledge about these resistance mechanisms.
The adopted methods are innovative and consistent with the aims of the research.
Results could constitute a great reference point for future research in the field.
Tables are comprehensive.
If authors will take into account the following recommendations, the quality of the work could be improved:
• In the ‘Introduction section’:
o The authors use sometimes the crop common name, sometimes the scientific one. Please make the text homogeneous.
o It could be suitable to highlight the new insights provided comparing them with the previous research studies.
o Furthermore, the authors could cite some other recent research study about genes involved in potyvirus-resistant watermelon response, such as that available at
https://www.tandfonline.com/doi/full/10.1080/07060661.2021.2021450#abstract
· In the ‘Results’ section:
o some parts of Figure 2 (fig2.A, fig2E and fig2F) are not readable. It could be appropriate to create separate figures.
o Some introductive considerations are not suitable for this section, such as those at Lines 123-126, 160-162. I suggest to remove these sentences or to move them in the ‘Introduction’ section or in other sections.
· In the ‘Discussion’ section, some sentences are more suitable for the ‘Introduction’ section.
· The ‘Conclusions’ section is absent, as well as the description of the impact of the achieved results on future research and/or development programs in the field.
· A carefully rereading of the manuscript is strongly recommended, mainly to improve and better organize the sentences in the manuscript.
· Some minor issues:
o Please check the Italic form of the mentioned genes – Lines 96, 97, 237, 241, etc
Comments on the Quality of English LanguageMinor editing of English language required.
Author Response
For research article
Response to Reviewer 2 Comments |
||
1. Summary |
|
|
Thank you for arranging a timely review of our manuscript. We have carefully considered the reviewers’ critical comments and modified the manuscript according to the reviewers’ thoughtful suggestions. A new version of our manuscript was also submitted. Please find the detailed responses below. Reviewer comments are pasted with bold font and we respond to each point made during the reviews in blue font. All the changes in the revised manuscript were highlighted in red. |
||
2. Questions for General Evaluation |
Reviewer’s Evaluation |
Response and Revisions |
Does the introduction provide sufficient background and include all relevant references? |
Yes/Can be improved/Must be improved/Not applicable |
Thank you very much for the suggestion. All the point-to-point response were listed in detail as below |
Is the research design appropriate? |
Yes/Can be improved/Must be improved/Not applicable |
|
Are the methods adequately described? |
Yes/Can be improved/Must be improved/Not applicable |
|
Are the results clearly presented? |
Yes/Can be improved/Must be improved/Not applicable |
|
Are the conclusions supported by the results? |
Yes/Can be improved/Must be improved/Not applicable |
|
3. Point-by-point response to Comments and Suggestions for Authors |
Comments 2: Overall, the work allows to improve our knowledge about these resistance mechanisms. The adopted methods are innovative and consistent with the aims of the research. Results could constitute a great reference point for future research in the field. Tables are comprehensive. If authors will take into account the following recommendations, the quality of the work could be improved:
Response: Thank you for your critical comments and suggestions. We have modified the manuscript and the Tables was changed as bellow:
Gene Name |
Gene ID |
CDS Length(bp) |
Protein Length(aa) |
Molecular weight |
CleIF4E1 |
Cla97C03G058500 |
708 |
235 |
26537.33 |
CleIF4E2 |
Cla97C06G116990 |
804 |
267 |
30284.48 |
CleIF4E3 |
Cla97C10G196220 |
2253 |
750 |
84106.20 |
CleIF(iso)4E |
Cla97C03G063210 |
2685 |
894 |
101536.21 |
- In the ‘Introduction section’:
o The authors use sometimes the crop common name, sometimes the scientific one. Please make the text homogeneous.
——Response: Thank you for your suggestions. We checked that and changed the scientific names into the common name.
Durable and effective resistance due to the inactivation of eIF4E has been achieved in the model plant Arabidopsis [11] as well as in a wide range of crops, including tomato [12-14], Chinese cabbage [15,16], tobacco [17,18], wheat [19,20], potato [21], and beet [22].
Examples include Chinese cabbage [16] infected with the turnip mosaic virus and beet infected with the beet mild yellowing virus [22].
o It could be suitable to highlight the new insights provided comparing them with the previous research studies.
——Response: Thank you for your suggestions. We have highlighted the new insights in the last introduction part.
In this study, the watermelon genome was screened to identify eIF4E family members. Additionally, the main CleIF4E gene was edited by CRISPR/Cas9 technology and non-transgenic-homozygous mutants (T2 generation) were obtained. This mutant not only exhibits defects on plant morphology, but also on male fertility ability. Furthermore, the mutant were resistant to infections of zucchini yellow mosaic virus (ZYMV), but not the cucumber green mottled mosaic virus (CGMMV). The roles of other CleIF4E genes were also explored in CleIF4E1 mutants and viral-infected mutants, revealing that CleIF4E1 plays a critical role in plant development and also viral resistance.
o Furthermore, the authors could cite some other recent research study about genes involved in potyvirus-resistant watermelon response, such as that available at
https://www.tandfonline.com/doi/full/10.1080/07060661.2021.2021450#abstract
——Response: Thank you for your suggestions. We have added in the last discussion part.
As the adverse effects of eIF4E mutations on fertility in watermelon, tissue-specific editing of these genes using a precise CRISPR-based technique may mitigate this influence. In addition, other reported resistant genes should be noted, including the ribosome-inactivating protein (RIP) genes found in watermelon [43] and the eukaryotic translation initiation factor interacting factors-EXA1 in Arabidopsis thaliana[44].
- Chanda, B.; Wu, S.; Fei, Z.; Ling, K.-S.; Levi, A. Elevated expression of ribosome-inactivating protein (RIP) genes in potyvirus-resistant watermelon in response to viral infection. Can. J. Plant. Pathol. 2022, 44, 615-625.[CrossRef].
- Nishikawa, M.; Katsu, K.; Koinuma, H.; Hashimoto, M.; Neriya, Y.; Matsuyama, J.; Yamamoto, T.; Suzuki, M.; Matsumoto, O.; Matsui, H.; et al. Interaction of EXA1 and eIF4E family members facilitates potexvirus infection in Arabidopsis thaliana. J. Virol. 2023, 97,1-16.[CrossRef]
- In the ‘Results’ section:
o some parts of Figure 2 (fig2.A, fig2E and fig2F) are not readable. It could be appropriate to create separate figures.
——Response: Thank you for your suggestions. We have simplified the Figure 2.
Figure 2. Editing of CleIF4E1 in watermelon using CRISPR/Cas9 technology. (A) Process for selecting edited lines from the cotyledon to adult plants. (B) Genomic map and CleIF4E1 target sites. The protospacer adjacent motif (PAM) is indicated with underlined red letters. (C and D) Comparison of the eIF4E nucleotide sequence and amino acid sequence between wild-type (wt) and mutant lines. Red arrows indicate the Cas9 cleavage sites.
o Some introductive considerations are not suitable for this section, such as those at Lines 123-126, 160-162. I suggest to remove these sentences or to move them in the ‘Introduction’ section or in other sections.
——Response: Thank you for your suggestions. We have deleted Lines 123-126 and this part was changed as below:
2.3 Effect of knocking out CleIF4E1 using CRISPR/Cas9 technology
Inactivating eIF4E may differentially affect diverse plants[7]. In the current study, we analyzed the effects of knocking out CleIF4E1 on watermelon plant growth and development. The comparison with wild-type (WT) plants indicated that inactivating CleIF4E1 in watermelon results in weak developmental defects. At the seedling stage, the mutant exhibited weak growth and the leaves were relatively small and wrinkled (Figure 3A). During the subsequent growth stages, the mutant plant exhibited dwarfism, but the leaf wrinkling symptom gradually weakened (Figure 3A). Moreover, there were changes to reproductive ability. Specifically, the fruit of the mutant line was smaller than that of the WT control (Figure 3C). After harvest, only a few seeds were obtained from the mutant; these seeds were smaller than the WT seeds, which may have contributed to the decrease in the mutant seed weight (Figure 3D, E and F).
The expression levels of CleIF4E2, CleIF4E3, and CleIF(iso)4E were also detected with a
quantitative reverse transcription polymerase chain reaction (qRT-PCR) method. Results showed that the relative expression levels of other eIF4E family genes was not increased in the CleIF4E1 knockout mutant(Figure 3B).
We moved Lines 160-162 to the fourth paragraph of ‘Introduction’ part,
However, during cultivation, watermelon plants are susceptible to many viruses, especially potyviruses [31]. Such as ZYMV, which is a typical potyvirus, is the causative agent of a destructive disease of cucurbit crops. It seriously affects crop quality and yield, resulting in considerable economic losses worldwide [32].
- In the ‘Discussion’ section, some sentences are more suitable for the ‘Introduction’ section.
——Response: Thank you for your suggestions. We have moved some sentences in the ‘Discussion’ section into the Introduction’ section. Such as sentences in the first paragraph of ‘Discussion’ section was moved to the second paragraph of Introduction’ section in the revised manuscript.
As the eIF4E factors are essential for host endogenous translation process, inactivating eIF4E may differentially affect diverse plants[7]. In cucumber, disrupting eIF4E expression using CRISPR/Cas9 technology does not result in observable alterations to plant development, but the generated plants exhibit broad-spectrum resistance to many viruses (e.g., ZYMV, papaya ring spot mosaic virus-W, cucumber vein yellowing virus, and watermelon mosaic virus) [23,24]. This reflects the potential utility of eIF4E genes for developing disease-resistant cultivars. However, inactivating eIF4E genes in Cucumis melo has also been associated with obvious growth defects or abnormal plant fertility [25], which may limit the use of this gene for improving agricultural production.
- The ‘Conclusions’ section is absent, as well as the description of the impact of the achieved results on future research and/or development programs in the field.
——Response: Thank you for pointing this out. We have added the The ‘Conclusions’ section.
- Conclusions
In this study, four eIF4E family members was identified in the watermelon genome and the main CleIF4E gene was knocked out using CRISPR/Cas9 technology. The resulting non-transgenic mutants (T2 generation) exhibits developmental defects in plant growth and fertility. Furthermore, the mutant was protected against the zucchini yellow mosaic virus, but not the cucumber green mottled mosaic virus. The study findings may provide the basis for future research conducted to generate viral disease-resistant watermelon cultivars.
- A carefully rereading of the manuscript is strongly recommended, mainly to improve and better organize the sentences in the manuscript.
——Response: Thank you for your suggestions. We have carefully read the manuscript and improved the organization of the sentences in the revised manuscript.
- Some minor issues:
o Please check the Italic form of the mentioned genes – Lines 96, 97, 237, 241, etc
——Response: Thank you for your suggestions. We have changed this.
Lines 96, 97:
Thus, we first designed sgRNAs that specifically target CleIF4E1, but not CleIF4E2 or CleIF4E3, to ensure only CleIF4E1 was knocked out using CRISPR/Cas9 technology.
Lines 237, 241:
Next, CleIF4E1 was knocked out using CRISPR/Cas9 technology. We ultimately obtained a homozygous mutant that contained an 86 bp deletion in the CleIF4E1 coding region, leading to a frame-shift in the coding region and the premature termination of CleIF4E1 expression (Figure 2C and D).
- Comments on the Quality of English Language: Minor editing of English language required.
——Response: Thank you for your suggestions. We have carefully revised the manuscript and also edited the English with the help of editing service. Certification is attached.

Round 2
Reviewer 1 Report
Comments and Suggestions for Authors
Authors have addressed my concerns.